

# Hail storm hazard in urban areas: identification and probability of occurrence by using a single-polarization X-band weather radar

Vincenzo Capozzi[1], Errico Picciotti[2], Vincenzo Mazzarella[1], Giorgio Budillon[1], Frank Silvio Marzano[2,3]

[1] Department of Science and Technology, University of Naples "Parthenope", Centro Direzionale di Napoli, 80143, Italy
[2] Centre of Excellence CETEMPS, University of L'Aquila, 67100, Italy
[3] Dept. of Information Engineering, Sapienza University of Rome, 00184, Italy

*Correspondence to*: V. Capozzi (vincenzo.capozzi@uniparthenope.it)

**Abstract.** This work exploits the potentiality of hail warning, based on single-polarization X-band weather radar measurements and tested on a large and well-documented data set of thunderstorm events in southern Italy near Naples. Even though X-band radars may suffer of two-way path attenuation especially at long ranges, due to their relatively low cost their use is rapidly increasing for short-range applications such as urban environments. To identify hail through radar measurements, two different methodologies have been selected and adapted to X-band data within the study area: one uses the Waldvogel (WAL) approach, whereas the other one uses the Vertically-Integrated Liquid Density (VIL-Density) product. The study aims at developing a Probability-of-Hail (POH) index in order to support hail risk management at urban scales. In order to find the optimal threshold values to discriminate between hail and severe rain, an extensive intercomparison between outcomes of the two methodologies and ground truth observations of hail has been performed, using a 2x2 contingency table and statistical scores.

The results show that both methods are accurate for hail detection in the area of interest, although VIL-Density product is less satisfactory than WAL method in terms of false alarm ratio. The relationship between the output of these two methodologies and POH has been derived through a heuristic approach, using a third-order polynomial fitting curve. As an example, the POH indexes have been applied for the thunderstorm event occurred on 21 July 2014, proving to be reliable for hail core detection.

## 1 Introduction

In recent years, urban areas have experienced a strong vulnerability to severe convective events, such as torrential rainfall, hail, lightning, wind storms and convective outbreaks (Marzano et al., 2012). The increased exposure to meteorological hazards is due both to the climate change, which has altered the temporal and spatial patterns of precipitation, and to the diffusion of complex networks infrastructures. Because of the high cost associated with severe weather occurrence, the need in urban areas of reliable monitoring and nowcasting techniques of weather hazards has recently increased. Nowdays, most of those techniques are based on weather radars, because the latter are capable to monitor precipitation at a resolution and areal extent previously impossible with rain gauge networks. The use of weather radar in urban hydrology is mainly focused on the development of high-resolution rainfall estimates, which serve as input for the modeling of urban drainage systems (e.g. Tilford et al., 2002).



In this work, we explore the usefulness of meteorological radar in urban environment for the real time detection of hail precipitation, which is one of the most widespread and damaging convective hazard. Hail events cannot be easily monitored using ground observational networks, due to their high variability in time and space. The only instrument that is able to measure hail size and kinetic energy distributions is the disdrometer (a sensor that measures particle size distribution). However, the

high costs and fragile nature of this instrument prohibit the growth of a dense network of stations. Therefore, as stated in Kunkel et al. (2013), hail (and all severe convective hazards, including tornadoes and thunderstorm winds) was considered in the lowest category of detection knowledge. The majority of the available information on hail and its frequency has come from hail storm reports provided by volunteers. Those reports have the drawback of being subjected to large spatial and reporting biases resulting from population density.

For the reasons stated above, weather radars appear as a valuable instrument for the real time observation of hail. In addition, volumetric radar data allow estimating the vertical extension of thunderstorm cells. The latter information is very useful for hail detection purpose, since the severity of thunderstorm is strongly related to its vertical extension (Delobbe et al., 2003).

Nowadays, the methodologies based on dual polarization weather radar are the most used to distinguish between rain and hail (e.g. Aydin et al., 1986; Straka et al., 2000; Marzano et al., 2007), because they are able to provide a direct distinction

between the rain droplets and the hailstones. As well as measuring the standard horizontally-polarized reflectivity ($Z_H$), a dual polarization weather radar also measures the ratio of the reflected horizontal and vertical power returns ($Z_V$). This ratio, called ZDR (differential reflectivity), usually in pure rain is positive (especially for heavy rain) whereas in hail its value is near zero (the latter tends to tumble randomly and thus, to the radar, appears to be spherical).

Single-polarization weather radars cannot distinguish among different types of hydrometeors. However, some

features in the vertical profile of reflectivity, including high reflectivities occurrences above specific environmental temperature heights, can represent the physical processes related to the hail growth (Skripnikovά and Řezáčová, 2014). Several methodologies for the identification of hail events from single-polarization weather radar measurements have been proposed in the available literature (Holleman, 2001). Some methods use threshold values for a hail-related quantity, such as reflectivity in low-level CAPPI (Geotis, 1963), vertically-integrated liquid density (Amburn and Wolf, 1997) or the difference in altitude

between the height of zero-degree isotherm and the highest level attained by 45-dBZ reflectivity core (Waldvogel et al., 1979). Other methods transform radar-based products into probability values, such as the Hail Detection Algorithm (Witt et al., 1998) and the Probability of Hail (Delobbe and Holleman, 2006). Several algorithms combine radar measurements with other meteorological data; for example, aerological data are used in Waldvogel et al. (1979) and in Witt et al. (1998). Auer (1994) proposed a technique that matches radar reflectivity measurements with satellite-derived cloud top temperatures. Various hail

identification techniques for Ebro valley region in Spain have been considered in Mallafre et al. (2009): among these, the one based on kinetic energy flux was found to be the best for distinguish between hail and no-hail precipitation in the studied area.

However, the charges of installation and maintenance of conventional weather radars, such as S-band and C-band systems, constitute one of the major limitation for their widespread diffusion. A good and reliable alternative is represented by low-cost short-range X-band radar systems. Although the latter suffer strongly from beam attenuation along the path, especially





in convective rain events and at long ranges, they have the advantage of performances comparable with conventional systems, but typically with smaller antenna and lower power consumption (Van de Beck et al., 2010). The impact of two-way path attenuation at X band is reduced if these low-cost radars are used at short ranges, as for urban area monitoring. In recent years, the increasingly use of X-band weather radars, both in single and in dual polarization mode, encouraged the development of

systematic approaches to mitigate their limitations and to enhance their promising potential in severe weather surveillance at short ranges (e.g. Anagnostou et al., 2004; Matrosov et al., 2005; Montopoli et al., 2010; Marzano et al., 2010; Van de Beek et al., 2010; Marzano et al. 2012).

Since November 2011, a single-polarization X-band weather radar, called WR-10X, has been installed in Naples' urban area at the top of Castel Sant'Elmo (40.8438°N, 14.2385°E, 280 m asl). The radar belongs to the Campania Center for

Marine and Atmospheric Monitoring and Modelling (CCMMMA) of University of Naples "Parthenope". First studies have been devoted to quality control and correction of radar data, to improve WR-10X quantitative precipitation estimates with a raingauge-based adjustment method (Capozzi et al., 2014) and to generate a preliminary hail detection product based on vertically-integrated liquid density method (Marzano et al., 2012; Capozzi et al., 2015).

The aim of this work is to develop a probability of hail occurrence algorithm, based on WR-10X measurements, that

could be particularly useful to mitigate hail-induced risks in Naples urban area. In order to achieve the goal of this study, two hail detection methods were applied and adapted: the Waldvogel criterion (WAL) (Waldvogel et al., 1979) and the Vertically-Integrated Liquid Density (VIL-Density) technique (Amburn and Wolf, 1997). The first one combines radar reflectivity data with other sources of information (sounding data or meteorological in situ data), whereas the VIL-Density product is derived exclusively from radar scans. Both methods have been selected and adapted to X-band after considering their easy exploitation

for an operational use.

The originality of this paper, with respect to previous studies devoted to the development of a radar-based hail detection product, lies in the two following aspects:

1. For the first time, using a large dataset of thunderstorm events, a single-polarization X-band miniradar has been

used to generate an innovative experimental probability of hail index valid for an urban area. Therefore, the advantages introduced by X-band systems in terms of better spatial and temporal resolution than conventional weather radars are fully exploited. The prototype radar system tested in this study constitutes one of the challenges in the near future radar meteorology, because it combines a relatively high quality of measurements with affordable cost of infrastructure.


2. With respect to the first preliminary studies (Marzano et al., 2012; Capozzi et al., 2015) performed about the topic addressed in this paper, a technique that matches radar reflectivity data with other sources of meteorological information, i.e. the WAL method, has been used to generate a probability of hail index. Therefore, the potential advantages introduced by the combination between single-polarization X-band miniradar measurements and





conventional meteorological data are for the first time explored. In addition, we optimize the performance of WAL method in relation to the study area and to X-band miniradar tested in this work, by proposing a variant of the original criterion developed by Waldvogel et al. (1979).

The two different algorithms (WAL and VIL-Density) have been compared and verified for 53 thunderstorm events observed in Campania Region between April 2012 and June 2015. Days with thunderstorm occurrence in the examined period have been identified means of synop stations, while hail reports have been provided by weather amateurs and newspapers. The evaluation of the two hail detection techniques is based on a number of verification scores (probability of detection, false alarm ratio, critical success index and probability of hail), that are calculated from contingency table.

The hail detection product developed in this work could be of high interest in the context of operational meteorology. Although it was developed for a specific area (i.e. Naples metropolitan environment), it can be easily adapted to other specific areas where detailed meteorological information is needed for a solid severe weather surveillance system.

This paper is organized as follows. In section 2, the two selected algorithms for hail detection are described. The third section provides information about radar data processing and a description of the study area; this section also addresses the hail events examined and the verification scores used in statistical analysis. The results of the study are presented in the fourth section. Finally, in the section 5 the capabilities of the two criteria are applied to a summer hail event occurred in July 2014.

## 2 Hail detection radar algorithms

The two hail detection algorithms are here illustrated and their adaptation to Naples X-band single-polarization radar measurements discussed.

### 2.1 Waldvogel method

The WAL technique relies on the difference ($\Delta H$), in (km), between the maximum altitude attained by a determined reflectivity core ($H_{Zt}$) and the zero-degree isotherm height ($H_{T0}$) (Fig. 1).

$$\Delta H = H_{Zt} - H_{T0} \tag{1}$$

In the original criterion defined by Waldvogel et al. (1979), Eq. (1) has been computed using the 45-dBZ reflectivity core. The occurrence of hail is likely if:

$$H_{Z45} \geq H_{T0} + 1.4 \text{ km} \tag{2}$$



As highlighted from Fig. 2, the probability of hail rises as the altitude of 45-dBZ reflectivity core increases above the zero-degree isotherm level. In order to detect hail, this technique considers two elements involved in the hail formation process: the occurrence of a remarkable updraft (the altitude of the 45-dBZ echo) and the presence of a substantial amount of supercooled water droplets and/or ice (the altitude of a strong reflectivity core above the zero-degree isotherm level).

Among the methods that combine radar measurements with other sources of meteorological data, the WAL method has the advantage of a relatively simple implementation. However, it should be noted that errors related to the height assigned to the measured reflectivities could have a negative influence on the performance of WAL algorithm. These errors result from radar beam size and from the limited number of elevation scans. Sampling errors can determine both an underestimation and an overestimation of the highest level at which a determined reflectivity value is observed. As an example, the presence, at long
range, of large vertical reflectivity gradients may result in a relevant overestimation (Delobbe and Holleman, 2006).

The present study aims to find not only the best threshold value for the height difference (i.e. a threshold that properly distinguishes between hail and severe rain), but also the reflectivity core where the WAL method exhibits the best performances. In this respect, we applied WAL criterion computing Eq. (1) not only using $H_{Z45}$, but also as a difference between the maximum altitude attained by 35 and 40-dBZ reflectivity cores and $H_{T0}$.

15 .

## 2.2 Vertically-integrated liquid density method

The Vertically Integrated Liquid (VIL) was proposed as a proxy of storm cell severity by Greene and Clark (1972). The aim of the VIL product is to give an instantaneous estimate of the water content residing in an atmospheric layer. VIL converts
reflectivity data ($Z$), which are the direct measurement of a weather radar, into equivalent liquid water content ($M$), through the semi-empirical relation between $M$ (kg m$^{-3}$) and $Z$ (mm$^6$ m$^{-3}$):

$$M = 3.44 \cdot 10^{-6} \, Z^{4/7} \tag{3}$$

Thereafter, the liquid water content is integrated vertically:

$$\text{VIL} \equiv \int_0^{H_{top}} M \, \text{dh} = 3.44 \cdot 10^{-6} \int_0^{H_{top}} Z^{4/7} \, \text{dh} \tag{4}$$

and has units of kilogram per square meter (kg m$^{-2}$). $H_{top}$ is the maximum height of vertical layer, which depends on the distance
from radar and elevation scan strategies.

When calculating VIL, several factors have to be taken into account. VIL values in storm located too close to the radar site are underestimate, because the radar is not able of scanning high enough to reach the upper portions of the





thunderstorms (Brimelow et al., 2004). Problems are also encountered when making VIL measurements of storms occurring too far away from radar, because the relatively broad beamwidth of the radar could introduce uncertainties in the vertical resolution of the reflectivity data. Moreover, as highlighted in Delobbe and Holleman (2006), for highly tilted storms, the VIL product may be an unreliable indicator of the thunderstorm intensity. In this respect, a cell-based method that follows the cell

vertically has been proposed in Stumpf et al. (2004). This approach could provide better assessment of VIL for tilted storms than the grid-based method (Brimelow el al., 2004). In the latter, before vertically integrating, the amount of liquid water from each elevation scan are mapped into a predetermined Cartesian grid.

To overcome some inherent problems with using VIL alone, Amburn and Wolf (1997) proposed the Vertically-Integrated Liquid Density (VIL-Density) radar product as hail indicator. VIL-Density can be obtained dividing the VIL by

EchoTOP, which is the altitude (expressed in meters) of the highest bin measured by radar. The quotient is multiplied by 1000 in order to yield the results in $(g\ m^{-3})$:

$$\text{VIL-Density} = \frac{\text{VIL}}{\text{EchoTOP}}\ 1000 \tag{5}$$

VIL-Density can be useful for detect thunderstorms characterized by high reflectivities relative to their altitude; therefore, VIL-Density correlates well with storm cell containing hail cores. It is important to point out that VIL-Density only identify hail precipitation at high altitudes, since weather radar are not able to observe hail occurring on the ground. This problem may occur in cases of quite high freezing levels or if the hail encounters significant liquid water when falling.

**3 Radar data and hail events**

This section is aimed at describing the available X-band radar dataset and the selected hail events by showing the processing data chain and the variability of the considered meteorological scenarios.

**3.1 Radar data**

WR-10X is a single-polarization radar manufactured by ELDES srl. In operative mode, WR-10X performs a volume scan with

azimuth and range resolution of 3.0 degrees and 0.3 km, respectively, using a pulse repetition frequency of 800 Hz. WR-10X installation facility in Naples Castel Sant'Elmo is given in Fig. 3: a radome protects the parabolic type dish antenna, whose rotation rate is 20°/ second. The time for a full volume scan is about 3 minutes and the maximum available range is 72 km. The operational volume observation strategy, repeated every 10 minutes, includes 6 PPI sweeps, with antenna elevation angle ranging from 1 to 10 degrees.





Before any operational use, the raw radar reflectivity data have to be processed through a quality control chain. The latter is needed to mitigate some errors that affect the quality of radar measurements. Processing chain has been focused on some systematic errors that include biases and range-dependent effects, as ground clutter residuals, backscatter from sea surface, beam blocking by surrounding topography and beam attenuation along the path (Capozzi et al., 2014). The latter could

be important in heavy rain or hail at X-band (Marzano et al., 2003).

In order to implement the WAL method, ΔH has been determined using the maximum height of 45-dBZ reflectivity core, according to Eq. (1). From corrected radar volumes, the latter has been computed for each azimuth and range as the altitude of the highest radar beam where a reflectivity at least equal to 45-dBZ is measured. In the same manner, ΔH has been computed also for the 35 and 40-dBZ reflectivity cores. The freezing level has been derived from a linear interpolation of the

radiosonde data acquired at Pratica di Mare station (41.67°N, 12.45°E), located at a distance of 150 km from the WR-10X site. In order to assess the representativeness of this station, a systematic comparison with the height of freezing level calculated from temperature data measured in Naples has been performed. From in situ measurements, the vertical temperature profile has been determined assuming a temperature decrease with height at the average lapse rate of 6.5°C per kilometer. Radiosoundings collected at Pratica di Mare proved to be very reliable in estimating the altitude of zero-degree isotherm level

in the area of interest.

In order to implement the VIL-Density method, corrected reflectivity data Z have been converted into liquid water content (M) according to Eq. (3). In the next step, liquid water content has been integrated over each vertical column, using Eq. (4), to find the VIL product. Finally, VIL-Density has been calculated through Eq. (5), using VIL values and EchoTOP heights. For each radar training dataset, the radar polar volumes have been resampled onto a uniform Cartesian grid with 1-

km horizontal resolution and 0.5-km vertical resolution before the integration for VIL calculation.

## 3.2 Study area and hail events

The area covered by radar measurements includes the western sector of Campania Region, as highlighted by Fig. 4. In this area, thunderstorm events occur all year round; in the warm season (May to October), convection is mainly triggered by the

interaction between small-scale mechanism and synoptic-scale flow, while in cold season (November to April) it is associated with the passage of a cold front or with a cold occlusion. The analyzed region is bounded on the north by Matese mountains and on the east by Campania Apennine, which have height values of more than 2000 and 1500 m, respectively. These two orographic features constitute an important factor in the mesoscale meteorology of the region, enhancing convection systems development, especially in summer season. Sea-air interactions also play a crucial role in storm activity: in the early part of

fall season, when sea surface temperatures reach the maximum value ($\cong 26°C$), coastal areas are sometimes affected by heavy thunderstorms, that are able to produce considerable rainfall accumulation (> 100 mm) in a relatively short time. During warm season, convection can be even triggered by the low-level convergence between diurnal sea-breeze and synoptic-scale flow,



when an unstable environment is present. This mechanism usually occurs in the inland sectors (at a distance of about 20-30 km from the coast) and, in combination with orographic forcing, can cause the development of hail producing thunderstorms.

The database used in this works includes 53 thunderstorms that occurred in the study area from April 2012 through June 2015. Other events have been discarded since thunderstorms were not well detected by the radar scans; this is due to
beam blocking at lower elevations, especially along volcano Vesuvio direction (i.e. in the southeastern sectors of the area of interest). On the other hand, for thunderstorms that took place within the cone of silence (that is, near to the radar site), WR-10X is not able to scan the upper part of the thunderstorm, thus WAL and VIL-Density methods cannot be applied effectively.

Hail was produced by 34 thunderstorms, while the remaining 19 produced heavy rain mixed with few hailstones or without hail at all. Days with thunderstorm and hail occurrence in the examined period have been identified by means three
synop stations, belonging to the network of "Meteorological Service of Military Italian Aeronautics", whose data are available on an hourly basis. Because of the restricted spatial extension of the majority of hail events, the synop observers reported only a small sample of the total number of events. Therefore, in order to complete the ground truth data, reports of hail events provided by volunteers and local papers have been taken into account. However, some locations and times listed on volunteers' reports may not be accurate and some spotters may not have given their exact latitude and longitude. Moreover, some hail
events may have occurred in places were spotters were not available.

To reduce spatial mismatching effects, the comparison between the outputs of two algorithms and in situ hail reports has been performed selecting radar pixels falling within 15-km distance from hail report location. Among these pixels, the one with highest ΔH value in Eq. (1) and the one with highest VIL-Density in Eq. (5) have been considered. In order to reduce also temporal mismatching effects, at both ends of the time window observation a tolerance of 10 minutes has been applied.

Table 1 and Table 2 show the distribution of the number of thunderstorm events, with and without hail, for given ranges of ΔH and VLD, respectively. As expected, the number of events without hail decreases as ΔH and VLD values increase. For ΔH values greater of 1.0 km, almost all producing-hail thunderstorms have been recorded, while for ΔH less than 1.0 km most of thunderstorm events observed did not cause hail. For VLD values greater of 3.5 g m$^{-3}$, only producing-hail thunderstorms have been observed, while for VLD less than 2.5 g m$^{-3}$ most of thunderstorm events recorded did not cause hail.

Seasonal analysis results (Table 3) highlight some of the thunderstorm features above described. Warm season (May to October) events are characterized, on average, by greater VIL, VLD and ΔH values than those found for cold season (November to April). Summer thunderstorms are often related to convection over land, which leads to a strong updraft and to a deep vertical extension of clouds. Moreover, during summer the orographic features become very important for initiating convection. The hot and dry air from the inland sectors rises in the presence of these mountains and, in concert with the sea
breeze, advects moisture over land, providing the prime conditions for a strong convective forcing. The temperature difference between the inland sectors and the coasts is at maximum during the summer, but it also significant in late spring and early autumn: therefore, in the latter the origin of the storms is quite similar. During the winter, the mechanism is reversed: the inland sectors are cold and there is a weaker temperature gradient. Consequently, the cell are sustained by weaker updraft and



usually develop over the sea surface. In warm season, because of the higher water vapor content in the atmosphere and the stronger updraft, thunderstorms can produce relatively large hail, which can be very dangerous. However, hail formation processes are strongly related not only with cloud height and updrafts magnitude, but also with the cloud depth below the freezing level (Pappas, 1962). In this respect, the lower freezing level found for winter cases can enhance the production of

hail stones, although smaller and less dangerous than those of warm season events, because a greater depth of the thunderstorm could have sub-freezing temperatures.

## 4 Methodology and results

This section is devoted to the brief description of the verification methodology, the set up of the proposed X-band technique for estimating the probability of hail and, as an example, the application of the latter to a recent hail event.

### 4.1 Verification scores

The comparison of the outcomes of the two selected methodologies and ground-truth hail reports was primarily concerned with finding a warning threshold that would correctly identify the hail cases with a minimum of false alarms. Thunderstorms

events have been classified using a 2-by-2 contingency table (Holleman, 2001). To determine the optimal threshold for hail alert in Naples urban area, the probability of detection (POD), the false alarm ratio (FAR), the critical success index (CSI) and the probability of hail (POH) were computed for some ΔH and VLD thresholds. The CSI summarizes the verification result in a single number and has been used to determine a good forecast decision threshold.

### 4.2 Statistical analysis and probability-of-hail index

Figure 5 shows, for the two selected methods, the behavior of the scoring indexes values (POD, FAR, CSI and POH) depending on their warning thresholds. For the WAL method, the results obtained using 35-dBZ (WAL 35-dBZ), 40-dBZ (WAL 40-dBZ) and 45-dBZ (WAL 45-dBZ) reflectivity cores to compute ΔH are presented. The behavior of scoring parameters are quite similar for the two algorithms. The results highlighted a lowering of both POD and FAR with increase of thresholds and

that CSI reaches the maximum value at a determined threshold. However, for WAL 35 and 45-dBZ a slight increase in FAR score has been observed for threshold values greater than 1.5 km. The highest CSI value (0.82) was obtained for the WAL 40-dBZ method. Using this reflectivity core, the CSI maximizes for ΔH = 1.0 km, which corresponds to a POD of 0.91 and to a FAR of 0.11. This ΔH threshold correctly detects as hail producing the 91% (31 of 34) of the thunderstorm events occurred in the examined period and falsely identifies as hail producing the 21% (4 of 19) of the thunderstorms without hail. For the WAL





45-dBZ method, CSI score maximizes for a height difference threshold (0.6 km) slightly lower than that carried out for the other two WAL criteria.

As concerns as VIL-Density method, the highest CSI (0.76) has been observed using a warning threshold of 2.4 g m$^{-3}$. This VLD threshold appears to be reasonable, identifying all of the hail producing thunderstorms. However, the FAR score carried out for this method (0.24) is higher than that found for the WAL 40-dBZ algorithm. The optimal VLD threshold for hail occurrence found in this study is much lower than those discovered in the previous studies (e.g. Amburn and Wolf, 1997; Rose and Troutman, 1997; Roeseler and Wood, 2001; Lahiff, 2005). The results obtained from these studies show that VLD threshold values which correctly identify the hail-producing storms are within a range between 3.1 and 3.7 g m$^{-3}$. However, the threshold values for a specific area can be affected by issues such as topography, height of the radar above sea level and the local climatic context. Therefore, the present work confirms that, in order to make an optimal use of VIL-Density, it is important to perform a local study aiming to find the best thresholds value for a determined area (Lahiff, 2005).

The analysis of cases study also allowed obtaining an empirical relationship that can be used in operational mode to convert $\Delta H$ (= $H_{Z40} - H_{T0}$) and VLD into POH. For both hail detection criteria outcomes, the relationship has been expressed with a third-order polynomial fitted curve:

$$POH_{WAL} = p_1\,(\Delta H)^3 + p_2\,(\Delta H)^2 + p_3\,(\Delta H) + p_4 \tag{6}$$

$$POH_{VLD} = c_1\,(VLD)^3 + c_2\,(VLD)^2 + c_3\,(VLD) + c_4 \tag{7}$$

The best fit between $\Delta H$ thresholds and POH is obtained when $p_1 = 0.1181$, $p_2 = -0.5291$, $p_3 = 0.7821$ and $p_4 = 0.4975$ (Fig. 6a). For VIL-Density method, the best fit is achieved when $c_1 = 0.009344$, $c_2 = -0.1106$, $c_3 = 0.5057$, $c_4 = 0.05351$ (Fig. 6b). For an operative use, for WAL method, when $POH_{WAL}$ is greater than 0.87 (87% in percentage), a hail event is occurring. In the same manner, for VIL-Density method, when $POH_{VLD}$ is above 0.76 (76%), hail precipitation is occurring.

### 4.3 Application to case study on 21 July 2014

A severe hailstorm occurred on 21 July 2014 in the Sorrentine Peninsula, a region that separates the Gulf of Naples to the north from the Gulf of Salerno to the south. The synoptic situation, very unusual for the summer season, was characterized by a low-pressure system over the northern Italy, affecting the Central Mediterranean basins (Fig. 7, upper panel). This low determined a cold air advection over the northern and central sectors of Italian peninsula and a warm and moist meridional flow over southern Tyrrhenian sea. Therefore, the study area was located along the boundary of the two adjacent air masses. This thermal boundary was associated with a low-level wind convergence line, with a jet-stream in the upper levels and was characterized by a strong advection of moisture. Those factors promoted a strong convective forcing, as is often observed for Mediterranean heavy precipitating events (Lebeaupin et al., 2006). Moreover, as usual in summer season, sea surface temperature was high (24°C) in the area of interest and favored evaporation, which supplied moisture to the low-levels as well as increasing the



convective instability. According to Pratica di Mare radio-sounding data collected at 12:00 UTC, the isotherm of zero degree was roughly 3500 m a.s.l. in the area of interest.

Images from satellite and WR-10X weather radar give useful information about the initiation and the evolution of the hailstorm. The latter developed in the early afternoon (at 14:30 UTC) offshore of Naples Gulf, along the thermal boundary

above described. According to Meteosat Second Generation visible (0.6 µm) image (Fig. 7, bottom panel), a V-shaped structure of the thunderstorm can be assumed at 15:30 UTC. The latter indicates the presence of a strong updraft, which was blocking the horizontal south-western flow: as the wind was forced around the updraft, it created a V-shaped structure, whose horizontal extension was around 120 km. Figure 8 displays a sequence of WR-10X images (showing Vertically Maximum Intensity, i.e. the maximum reflectivity value on the vertical of every point) that allows to obtain a time history of hailstorm displacement.

The thunderstorm cell showed its first clear signature on radar images at 14:35 UTC. In next 90 minutes, the cell rapidly moved from southwest to northeast, affecting the Gulf of Salerno and the southern sector of Naples Gulf. The convective system reached its peak intensity between 15:25 and 15:45 UTC, causing heavy rainfall and hail in Sorrento and Vico Equense cities. Within the area most affected by the hailstorm, there was one automatic rain-gauge, located in Sorrento city (40.62997 N, 14.38238 E). This rain-gauge recorded a total accumulation of 29.2 mm in one hour (from 15:00 to 16:00 UTC). The passage

of the thunderstorm core produced an amount of 16.8 mm in a ten-minutes period (15:40-15:50 UTC), with rain rates up to 200 m h$^{-1}$. According to local newspapers, hail precipitation lasted about 15 minutes, causing tangible damages to transport activities and crops. The hailstones had a diameter of up to 3-4 cm (Fig. 9), but no significant accumulation on the ground was observed.

When the hailstorm core passed over Sorrento city, it was characterized by $\Delta H$ (= $H_{Z40} - H_{T0}$) values up to 1.5 km

(Fig. 10a). Those values suggest that convective cell had a great vertical extension, that was favorable to the hail stones growth. In this respect, the presence of 40-dBZ echoes well above the zero degree level is a clear signature of a strong, wide and persistent updraft in the hail stones growth layer (-10°C to -30°C). In addition, VLD values up to 3.5 g m$^{-3}$ (Fig. 10c) also indicate the presence of a deep and intense hail core. Using Eq. (6) and Eq. (7), POH products for WAL 40-dBZ (Fig. 10b) and VIL-Density (Fig. 10d) methods have been generated, respectively. POH index, expressed as percentage, ranges from 0%

(no chance of hail) to 100% (certainty of hail). According to on-ground reports and observations, in the area of Sorrento a very high probability of hail ($\cong$ 88% for WAL 40-dBZ and $\cong$ 80% for VIL-Density) has been detected above the thresholds found in the previous paragraph. Therefore, in this case both POH indexes proved to be reliable in hail core identification.

## 5 Conclusions

The main aim of this study is to develop a hail detection algorithm, based on the measurements of a single-polarization X-

band weather radar, in order to provide a real-time hail warning in Naples metropolitan area. Based on reports provided by synop observers and other sources of information, 53 thunderstorm events occurred between April 2012 and June 2015 have collected with precise information about time and location. Among various methods proposed in literature, the Waldvogel





algorithm and the Vertically-Integrated Liquid Density method have been selected to identify hail from reflectivity data measured by X-band weather radar operating in Naples urban area. The first one relies on the difference (ΔH) between the maximum altitude attained by the 45-dBZ echo and the height of zero-degree isotherm. The capabilities of this method have been evaluated computing ΔH not only for the 45-dBZ reflectivity core, but also for the 35-dBZ and 40-dBZ cores. The output (VLD) of the second method is derived exclusively from radar scans and is obtained dividing the VIL, that is an estimate of the water content residing in an atmospheric layer, by the EchoTOP, the altitude of the highest bin measured by radar. The outcomes of the two radar-based hail detection criteria have been compared with ground truth observations, using a statistical scores technique based on a 2x2 contingency table. The verification analysis allowed obtaining ΔH and VLD thresholds that would reliably detect hail events with a minimum of false alarms and misses. For the Waldvogel method, the best results have been obtained for 40-dBZ reflectivity core: using this reflectivity value, the Critical Success Index (CSI), that summarizes the verification results in a single number, maximizes for ΔH = 1.0 km. For Vertically-Integrated Liquid Density method, the highest value for CSI was reached for VLD = 2.4 g m$^{-3}$. These thresholds have proven to be reliable in identifying the hail producing thunderstorms, detecting the 91% and 100% of the thunderstorms with hail occurred in examined period, respectively. However, from operational perspective, the FAR score found for Vertically-Integrated Liquid Density method (24%) looks quite high and should be improved.

The relationship between and the Probability of Hail (POH) and the output of the two hail detection criteria has been derived through a heuristic approach, using a third-order polynomial fitted curve. This relationship can be used in operational mode to convert ΔH and VLD values into a POH index, varying between 0 and 100%. An example of POH indexes performance has been shown for a hail event occurred on 21 July 2014 in the Sorrento Peninsula. In this case, both hail detection methodologies have proven to be reliable in hail core identification.

The results obtained from this study are very encouraging and can bring benefits for risk management associated to hail events. Future work shall be devoted to reduce the FAR score found for Vertically-Integrated Liquid Density; in this respect, more intercomparisons between radar and ground truth observations will be carried out to cover a larger number of events. Moreover for future analysis, in order to improve the hail detection products performance close to the radar site, the operational volume observation strategy will include more PPI sweeps at higher elevations angles.

**Data availability**

The WR-10X radar data used in this work can be accessed through the website of "Campania Center for Marine and Atmospheric Monitoring and Modelling" of the University of Naples "Parthenope" (meteo.uniparthenope.it).



Radiosonde data collected in Pratica di Mare (16245 LIRE) are provided by the Department of Atmospheric Science, University of Wyoming (weather.uwyo.edu/upperair/sounding.html).

The synop reports used to identify thunderstorm and hail occurrence in the study area can be accessed through the following website: www.ogimet.com/synops.phtml.en.

## Acknowledgments

The authors thank the "Soprintendenza Speciale per il Patrimonio Storico Artistico ed Etnoantropologico e per il Polo Museale della città di Napoli" for hosting the weather radar at Castel Sant'Elmo. ELDES srl (Florence, Italy) personnel is acknowledged for technical support. The authors are very grateful also to Campania Region Department of Civil Protection (DPC), Naples,
Italy, for having kindly granted the access to rainfall data acquired by their monitoring rain-gauge network.

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



**Table 1.** Number of thunderstorms with and without hail for given ranges of ΔH. The results obtained computing ΔH with the three different reflectivity cores used in this study (35, 40 and 45-dBZ) are presented.

| Events occurred in 2012-2015 period | Range in ΔH (km) | | | | | | | | | | | | | | |
|---|---|---|---|---|---|---|---|---|---|---|---|---|---|---|---|
| | <= 0.5 | | | 0.6 – 1.0 | | | 1.1– 1.5 | | | 1.6 – 2.0 | | | >= 2 | | |
| | 35 dBZ | 40 dBZ | 45 dBZ | 35 dBZ | 40 dBZ | 45 dBZ | 35 dBZ | 40 dBZ | 45 dBZ | 35 dBZ | 40 dBZ | 45 dBZ | 35 dBZ | 40 dBZ | 45 dBZ |
| Number of events with hail | 3 | 3 | 1 | 2 | 1 | 7 | 6 | 9 | 9 | 6 | 7 | 6 | 17 | 14 | 8 |
| Number of events without hail | 7 | 11 | 7 | 4 | 4 | 4 | 1 | 1 | 1 | 1 | 0 | 1 | 4 | 2 | 2 |

**Table 2.** Number of thunderstorms with and without hail for given ranges of VLD.

| Events occurred in 2012-2015 period | Range in VLD (g m$^{-3}$) | | | | |
|---|---|---|---|---|---|
| | < 2.5 | 2.5 – 3.5 | 3.6 – 4.0 | 4.1 – 4.5 | > 4.5 |
| Number of events with hail | 3 | 13 | 10 | 2 | 6 |
| Number of events without hail | 7 | 11 | 1 | 0 | 0 |



**Table 3.** Average values of some products generated by the two radar-based hail detection algorithms for warm season events (May to October) and for cold season events (November to April).

| Events occurred in 2012-2015 period | VIL (kg m$^{-2}$) | VLD (g m$^{-3}$) | $H_{T0}$ (m) | $H_{Z35}-H_{T0}$ (m) | $H_{Z40}-H_{T0}$ (m) | $H_{Z45}-H_{T0}$ (m) |
|---|---|---|---|---|---|---|
| Warm season events (May to October) | 9.5 | 3.6 | 3200 | 2700 | 2500 | 2200 |
| Cold season events (November to April) | 5.2 | 3.1 | 1900 | 1440 | 1300 | 1100 |



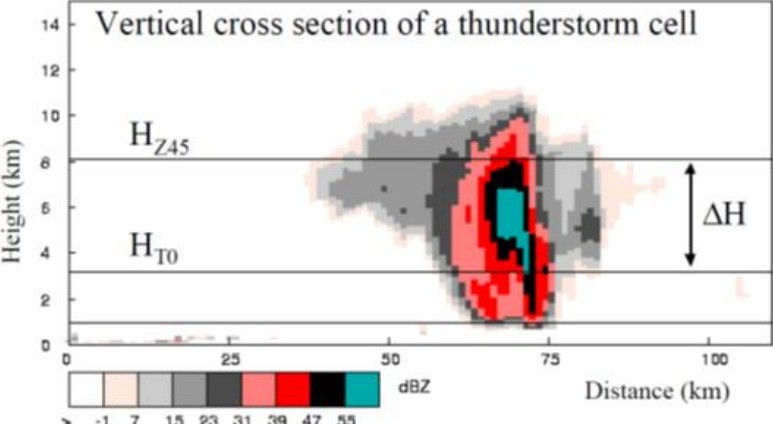

**Fig. 1.** Criterion of hail detection proposed in Waldvogel et al. (1979). The latter is based on the difference ($\Delta H$) between a radar height signature for strong updraft and large amounts of hydrometeors ($H_{Z45}$) and the freezing level ($H_{T0}$).

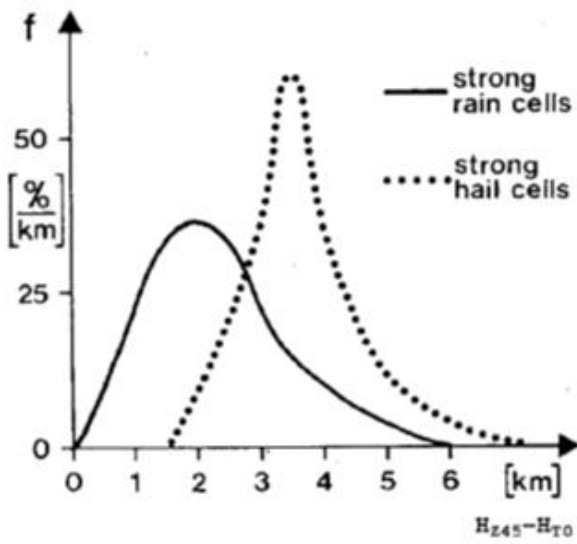

**Fig. 2.** Normalized frequency distributions of the parameter ($H_{Z45}-H_{T0}$) for strong rain cells and strong hail cells.



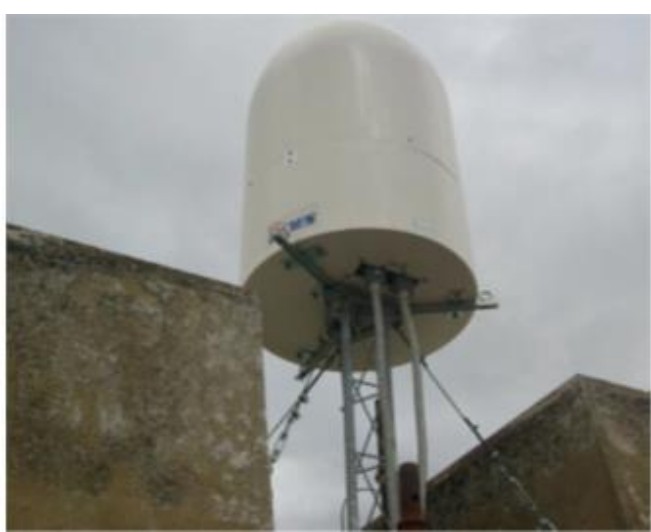

**Fig. 3.** WR-10X installation at Naples Castel Sant'Elmo site.

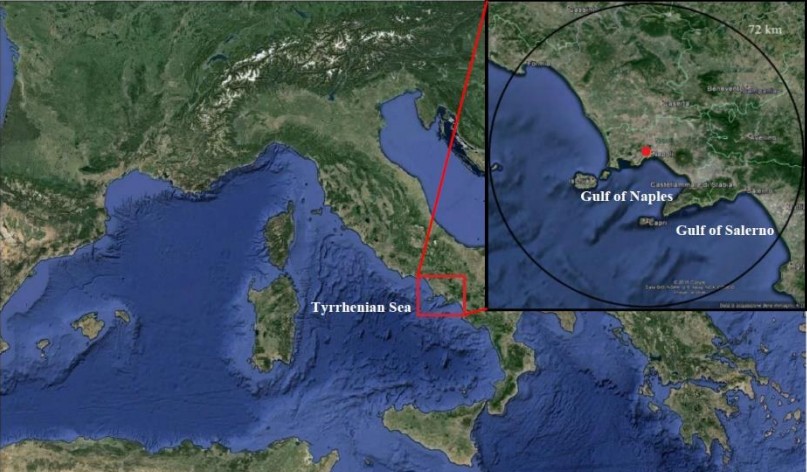

**Fig. 4.** Map of study region including radar location (filled-in red circle). Circular line at 72 km from Naples Castel Sant'Elmo indicates the limit of the area covered by radar.




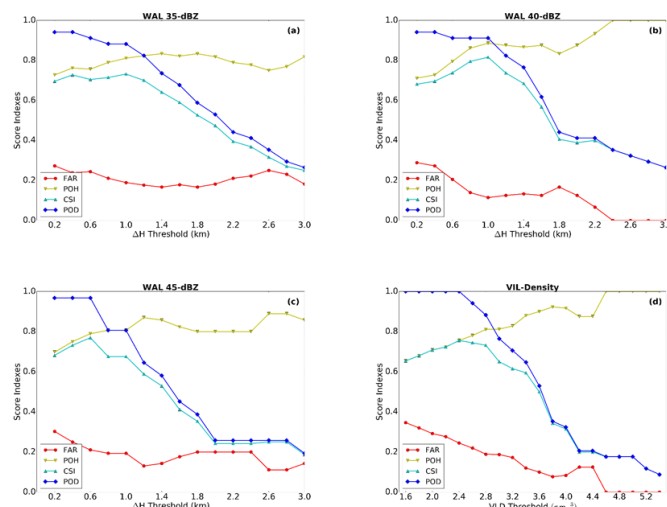

**Fig. 5.** The scoring parameters (FAR, POH, CSI and POD) for the two hail detection methods as a function of the warning threshold. In (a),
(b) and (c), the results obtained for the WAL method using the three reflectivity cores (35, 40 and 45 dBZ) tested in this study to compute
ΔH are presented. In (d) the results obtained for VIL-Density criterion are shown. The scoring parameters are deduced from the comparison
of the two methods with ground-truth verification data.

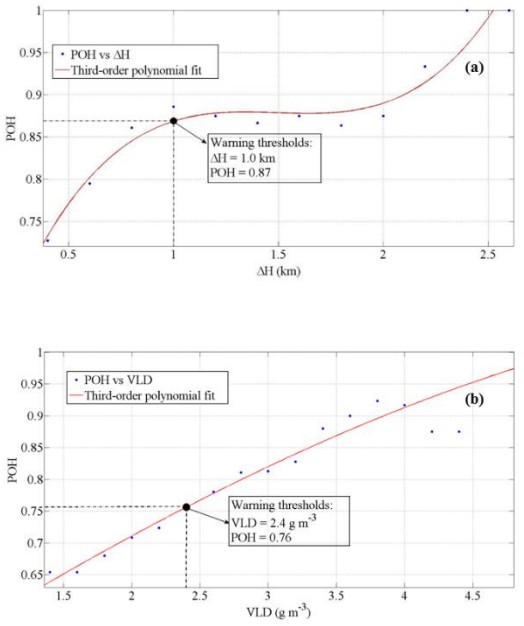

**Fig. 6.** The Probability of Hail (POH) as a function of ΔH (a) and VLD (b). The POH indexes, shown as red curves, have been obtained
through an empirical approach, using a third-order polynomial fit. The warning thresholds (i.e. the threshold above which hail is occurring)
are indicated as black filled circle over both panels.





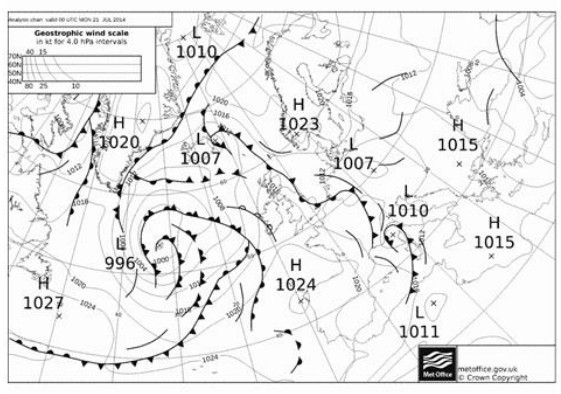 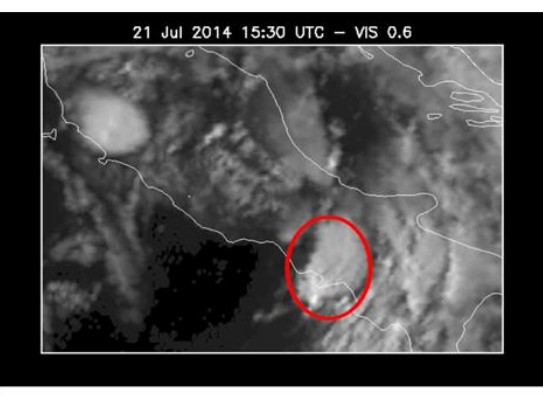

**Fig. 7.** In the left panel, surface pressure and weather fronts on 21 July 2014 (00:00 UTC) are shown. In the right panel, a Meteosat Second Generation visible (0.6 μm) image (collected at 15:30 UTC) is shown. The thunderstorm that caused hail precipitation in Sorrentine peninsula (highlighted by the red circle) exhibited a V-shaped pattern.

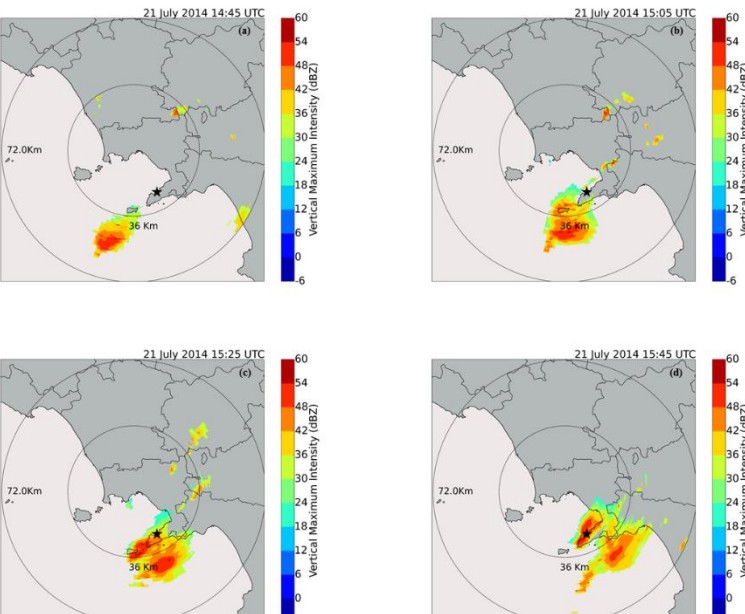

**Fig. 8.** WR-10X images showing the evolution of hailstorm that affected the city of Sorrento on 21 July 2014. Vertically Maximum Intensity (VMI) product obtained at 14:45 UTC (a), 15:05 UTC (b), 15:25 UTC (c) and 15:45 (d) is shown. The thunderstorm cell developed over Tyrrhenian sea and then moved from south-western to north-eastern, affecting both the Gulf of Naples and the Gulf of Salerno. Sorrento city
15    location is highlighted as black star.





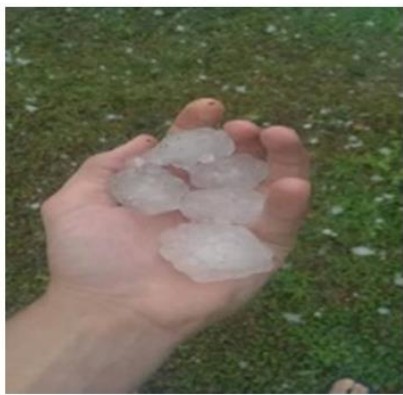
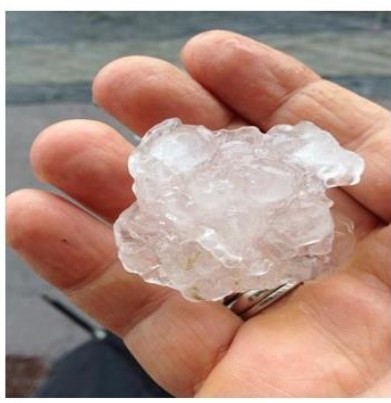

**Fig. 9.** Two photografic evidences of hailstones produced by convective cell that affected the city of Sorrento on 21 July 2014.

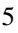

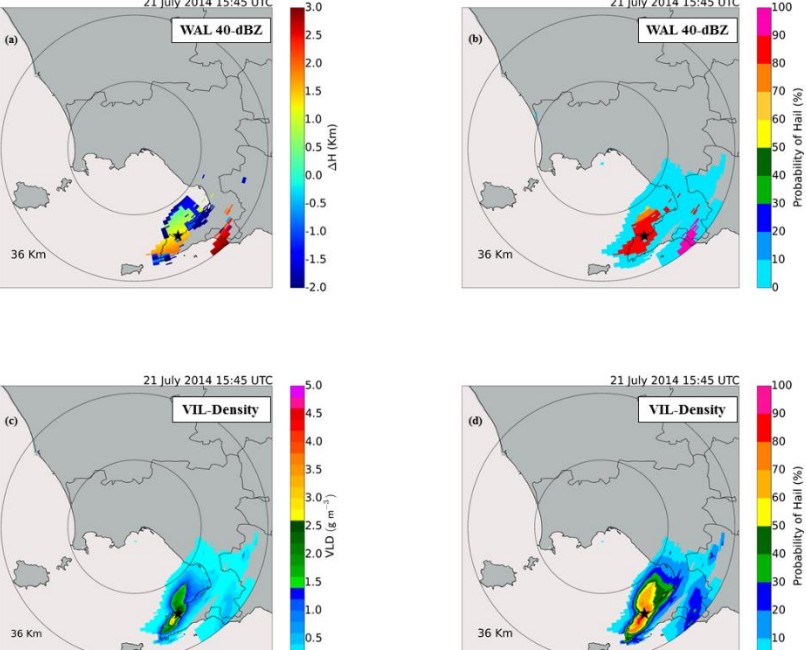

**Fig. 10.** Hail detection products and Probability of Hail indexes obtained by WR-10X when the hailstorm passed over Sorrento city (highlighted by black star). In the left panels, the ΔH map from WAL 40-dBZ criterion (a) and the VLD map from VIL-Density criterion (c) are shown. In (b) and (d), the related POH maps for the hail event occurred on 21 July 2014 (15:45 UTC) are presented. To better display the thunderstorm core features in the area of interest, a zoom on the Gulf of Naples has been performed.
