# Peer review of "Hail storm hazard in urban areas: identification and probability of occurrence by using a single-polarization X-band weather radar"

_Hydrology and Earth System Sciences, 2016_

## Referee Comment (RC1) · Anonymous Referee #1 · 23 Jul 2016

It is an interesting and well written paper. Please take into account the following:

specific comments: page 7 - line 2: describe in more details the error correction methodology page 9 - line 15: the content and use of the table is not clearly understood page 9 - line 16: POD, FAR, CSI. Describe in more details the methodology for their calculation page 12 - line 30: Are the WR-10X radar data available ?

technical corrections: page 7 - line 3: such as ground ... page 8 - line 3: works page 10 - line 3: concerning page 10 - line 12: cases study page 11 - line 16: rain rates up to 200 m h-1. it is better to express rainfall rate in mm/10min. page 12 - line 16: between and

---

## Referee Comment (RC2) · Anonymous Referee #2 · 25 Jul 2016

The paper describe the use of enstablished techniques to monitor and identify hail shaft in radar echoes. The novelty could be find in the use of such techniques at X band, even if it is well recognised that attenuation play an important role at such frequency when convective storm is on the beam path. It is stated that such problem is adressed in the processing chain, a reference is present, but some doubts are not confuted since the events considered are characterised by strong convective rainfall and hail. Some additional material to demonstrate that correction is effective or a deeper discussion of its effects is not relevant is needed.

As a general discussion about this type of paper one critical point is on how, when, where, how long, how intense and any other parameter related to hail fall could be

collected from volunteer reports (as well as from newspaper ) could be extracted and what is its quality and representativeness. I want to clarify that this is not a fault of the authors it is a simply a "matter of fact". Even synop messages from synop station could only report if hail falls just above the station.

In the following a list of specific comments is present.

1) 2.1 Waldvogel method.

In this section the Waldvogel method is presented. In such method an important role is played by the Ht0 variable. I suggest to move here (e.g. at the end of the section) the discussion on how such variable is computed. Does not make sense to have this discussion in the presentation of the radar (sect. 3.1).

It could be beneficial to add a discussion on the estimate of the error in the interpolation and the sensitivity of the algorithm to such error.

2) pag 8 - rows 8-15 Could be useful to report on a map where the events are located (hail reports).

3) pag 8 from row 25 to the end of section 3.2 The text in this section of paragraph 3.2 reports some convection initiation mechanisms, but since no previous work are referred neither there is an analysis that support such mechanisms I have to assume that this is an authors's inference. Since this is not relevant for the paper I suggest to remove it. Alternatively add a section to demostrate such processes.

4) pag 10 - row 14 Could you clarify what you have used as dependent variable and independent variable in the fitting curve? I assume that you use POH and H for Wal and POH and VLD for VLD. If this is correct could you explain why you need another way to calculate POH ? Is is related to error in POH estimate?

Please clarify.

5) Pag 10 - rows 21-22 Could you please explain how such threshold have been calculated?

6) Pag 11 row 5 Substitute "bottom panel" with "right panel"

7) pag 11 row 12 Add a marker for Vico Equense on the map.

8) 'pag 11 row 16 Adde reference to the local newspaper.

9) pag 11 row 32 There is no clear indication that hail information is "precise about time and localtion" Remodulate the sentence.

10) Table 3 Is mean the significative parameter for the distribution of the quantitites used? Add information about distributions (at least in the replay)

11) fig 7 Both panels are not clear, try to improve they.

12) Fig 8 and fig 10 Increase the level of zoom used. It is hard a good analysis from this picture. Is is not easy identify where hail is expected and where not.

This open a question on the identification of the hailfall location. Is the area, above threshold, much bigger that the area from the reports? If yes could you comment it?

---

## Author Comment (AC1) · 4 Aug 2016

The comment was uploaded in the form of a supplement:
http://www.hydrol-earth-syst-sci-discuss.net/hess-2016-177/hess-2016-177-AC1-supplement.zip

---

## Author Comment (AC2) · 4 Aug 2016

The comment was uploaded in the form of a supplement:
http://www.hydrol-earth-syst-sci-discuss.net/hess-2016-177/hess-2016-177-AC2-supplement.zip

---

## Referee Comment (RC3) · Anonymous Referee #1 · 8 Aug 2016

All required changes were made. The overall presentation is well structured and clear. The paper may be published as is.